# A database of marine macronutrient, temperature and salinity measurements made around the highly productive island of South Georgia, the Scotia Sea and the Antarctic Peninsula between 1980–2009

Michael J. Whitehouse, Katharine R. Hendry, Geraint A. Tarling, Sally E. Thorpe, Petra ten Hoopen

British Antarctic Survey, Natural Environment Research Council, High Cross, Madingley Road, Cambridge CB3 0ET, UK

*Correspondence to*: Petra ten Hoopen (peopen@bas.ac.uk)

**Abstract**. We present a database from substantial collections of macronutrient data made on 20 oceanographic cruises, primarily from around the island of South Georgia and the Scotia Sea. This sector of the Southern Ocean was studied comprehensively during the Discovery Investigations between ~1920 and 1950 and resulted in the hugely influential Discovery Reports. Following this pioneering research period, there was a lull of several decades prior to the British Antarctic Survey's (BAS) initiation of an offshore biological programme to study the ecology of the South Atlantic sector of the Southern Ocean. These studies began in the late 1970s and have continued until the present day. Between 1980-2009, the programme included macronutrient measurements as part of an integrated ecosystem analysis. In addition to South Georgia and the Scotia Sea, measurements were also made in the Bellingshausen Sea and the waters to the west of the Antarctic Peninsula. Data were collected during all months of the year with the exceptions of May and June and compiled into a database. Vertical profile samples were taken from water bottles while data along transects were collected through monitoring the ship's non-toxic seawater supply. Nutrients measured were silicate, $Si(OH)_4$-Si; phosphate, $PO_4$-P; nitrate, $NO_3$-N; ammonium $NH_4$-N; and nitrite, $NO_2$-N. Our database includes nutrient data along with contemporaneous temperature and salinity data where available. Further background and supporting information are included for context.

## 1 Introduction

### 1.1 Geography and physical oceanography

The island of South Georgia lies on the Scotia Ridge, a submarine arc in the southwest Atlantic sector of the Southern Ocean that extends eastwards from South America to the South Sandwich Islands, returning westwards to the Antarctic Peninsula. The arc forms the northern, eastern and southern boundary of the Scotia Sea with Drake Passage forming the western boundary.

South Georgia and its broad continental shelf lie within the Antarctic Zone (AAZ) of the predominantly eastward flowing Antarctic Circumpolar Current (ACC). The AAZ is bounded by the Polar Front (PF) to the north and the Southern Antarctic Circumpolar Current Front (SACCF) to the south (see Fig. 1; Nowlin and Klinck, 1986; Orsi et al., 1995). In the Scotia Sea, the ACC is deflected northeastward by the local bathymetry. The SACCF approaches South Georgia from the southwest and is inflected around the eastern and northern continental shelf of the island before retroflecting eastwards at the Northwest

Georgia Rise (Orsi et al., 1995; Thorpe et al., 2002; Meredith et al., 2003; Boehme et al., 2008). The oceanic circulation at South Georgia has been studied with Eulerian and Lagrangian observations, and modelled at increasingly higher resolutions. These studies demonstrate a northwestward flow along the island's northern shelf edge, cross-shelf exchange and areas of retention on the island's continental shelf (Hardy and Gunther, 1935; Maslennikov, 1979; Priddle et al., 1986; Latogursky et al., 1990; Trathan et al., 1997; Brandon et al., 1999, 2000; Young et al., 2014). Recent fine-scale ocean models not only

reconstruct the larger-scale circulation features but also resolve eddy-scale processes, and suggest the existence of coastal upwelling jet structures that have yet to be physically observed (Young et al., 2011, 2014; Matano et al., 2020).

The long-term mean positions of the ACC fronts mask the degree of large- and mesoscale variability. Meandering and eddy shedding have been described for ACC fronts (e.g., Lutjeharms and Baker, 1980; Bryden, 1983). In areas of irregular bottom

topography such as the northeastern Scotia Sea, the fronts are particularly variable (e.g., Gordon et al., 1977; Peterson and Whitworth, 1989; Boehme et al., 2008). The positions of fronts and eddies influence the waters and plankton communities around South Georgia. There is some evidence, for example, that eddies from north of the PF input warmer water to the island and introduce physical structure to the water column that facilitate the resupply of nutrients from deeper to surface waters (Atkinson et al., 1990; Whitehouse et al., 1996b).

**1.2 Research history**

South Georgia has been commercially exploited for centuries from the sealing of the late 1700s (Bonner, 1984; Headland, 1984) to the whaling industry in the early 20th century (Harmer, 1931; Kemp and Bennett, 1932) and the current krill fishery in the Southern Ocean (Everson and Goss, 1991; Trathan et al., 1998).

The Discovery Investigations were initiated in the 1920s to provide an ecosystem approach to managing whaling. Multi-ship surveys were undertaken to understand the causes of the region's high primary productivity and how it supported the fisheries. The resulting Discovery Reports covered many topics from oceanography to whales, and the report on the plankton by Hardy and Gunther (1935) linked the environment and higher trophic levels using some very modern concepts. For example, they suspected that micronutrients controlled phytoplankton productivity. Later Discovery Investigations broadened their coverage

to include South Georgia as part of the Scotia Sea-Antarctic Circumpolar Current system (e.g., Foxton, 1956; Marr, 1962; Mackintosh, 1973).

The Discovery Investigations laid the foundations for our understanding of phytoplankton growth and nutrient use at South Georgia. Hardy and Gunther (1935) and Hardy (1967) correlated locally reduced phosphate concentrations with elevated

phytoplankton biomass, suggesting that phosphate depletion provided a time-integrated "memory" of primary production. Clowes (1938) showed summer silicate and phosphate reductions, and possible year-to-year variation in phytoplankton utilisation, and suggested that silicate concentrations may, in some years, limit phytoplankton growth. A wider scale context was provided by Hart (1934, 1942) who noted the general failure of Antarctic marine phytoplankton to deplete fully the abundant pools of macronutrients in the surface waters: now known as the High-Nutrient-Low-Chlorophyll paradox (HNLC).


Between 1940 and 1970, much less scientific work was undertaken in the vicinity of South Georgia. However, from the late 1970s to the present-day, attention has been renewed with work being undertaken by the British Antarctic Survey (BAS). Altogether, offshore, ecological studies have been conducted sporadically in the South Georgia marine environment and in the Scotia Sea for about a century.

**1.3 Phytoplankton productivity and nutrient controls**

The AAZ's open ocean is generally considered to be a zone of low productivity. However, the South Georgia region is characterised by high biomass and productivity of phytoplankton, zooplankton and vertebrate predators. Although the AAZ is characterised by HNLC conditions (maximum chlorophyll $a$ ~1 mg m$^{-3}$), parts of the Scotia Sea may be more productive (but still usually <2.5 mg chlorophyll $a$ m$^{-3}$, Rönner et al., 1983; Jacques, 1989; Tréguer and Jacques, 1992; Korb et al., 2005).

However, during a series of cruises across the Scotia Sea, concentrations of chlorophyll $a$ >3 mg m$^{-3}$ were found at ~58°S in spring (Korb et al., 2012). Phytoplankton blooms >0.75 mg m$^{-3}$ chlorophyll $a$ persisted for >6 months during the 2006/2007 season in this mid-Scotia Sea zone. Similarly, a large-scale bloom investigated during the 2011/2012 season downstream of South Georgia revealed chlorophyll $a$ concentrations of >3 mg m$^{-3}$, peaking in December, with the bloom persisting (chlorophyll $a$ > 0.5 mg m$^{-3}$) until mid-March. Furthermore, in comparison to those in the open ocean, the South Georgia

bloom was characterised by higher chlorophyll $a$-specific carbon fixation, indicative of high photosynthetic efficiency (Hoppe et al., 2017).

With decades of both field observations and remotely sensed ocean colour data available to investigate spatial and temporal variability, there is clear evidence that phytoplankton blooms are more frequent and more intense downstream of South Georgia

and into the Georgia Basin relative to upstream (e.g., Borrione and Schlitzer, 2013). High phytoplankton concentrations (>20 mg chlorophyll $a$ m$^{-3}$) may be linked to enhanced supplies of iron (up to 4 nmol m$^{-3}$, de Baar et al., 1995; Whitehouse et al., 2000; Korb et al., 2004; Nielsdottir et al., 2012; Schlosser et al., 2018) or reduced forms of nitrogen (up to 3 mmol ammonium m$^{-3}$, Owens et al., 1991; Priddle et al., 1997; Whitehouse et al., 1999). Although macronutrients are generally non-limiting in the AAZ, silicate concentrations (<1 mmol m$^{-3}$) limit growth at South Georgia in some summers (Whitehouse et al., 1996a,

2008). During trans-Scotia Sea cruises from ~61-50°S, substantial latitudinal gradients were found for silicate, nitrate and phosphate (Whitehouse et al., 2012). Surface ammonium concentrations were substantially higher to the north of the North Scotia Ridge although a peak in values was measured around the southern boundary of the ACC, coincident with increased mid-Scotia Sea chlorophyll *a* concentrations. These transects crossed multiple oceanographic features that may each contribute to increased productivity and nutrient depletion. However, silicate was the only macronutrient to be found at limiting concentrations and only north of 56°S (Whitehouse et al., 2012).

Observations, together with coupled hydrodynamic and biogeochemical models, support a strong supply of iron downstream of South Georgia, sourced from shallow shelf sediments (<20 m water depth), glacial meltwaters and particulates, upwelling deep waters transported by the ACC and topographic steering, and a lesser contribution from dust (Nielsdottir et al., 2012; Borrione et al., 2014; Hoppe et al., 2017; Schlosser et al., 2018; Matano et al., 2020). The alleviation of iron limitation not only drives the phytoplankton bloom initiation, but also has the potential to shift the uptake ratio of macronutrients, impacting regional stoichiometry (Borrione et al., 2014; Hoppe et al., 2017). However, the lateral advection of iron alone cannot explain the variability in duration of downstream blooms near South Georgia (Robinson et al., 2016). Mixed-layer depths and phytoplankton-zooplankton interactions will also influence nutrient stoichiometry; grazing pressure plays an important role in bloom dynamics and rapid, shallow nutrient cycling, and effectively traps nutrients in the surface mixed layers of the downstream waters (Robinson et al., 2016; Schlosser et al., 2018; Cavan et al., 2019).

Although there is intensive recycling in the surface mixed layer, the high summer primary productivity to the north of South Georgia is reflected in organic carbon export and inorganic carbon uptake. For example, in austral summer 2012, the flux of particulate organic carbon (POC) collected in sediment traps downstream and upstream of South Georgia (1500-2000 m water depth) were 46-904 and 38-205 $\mu$mol m$^{-2}$ d$^{-1}$ respectively (Rembauville et al., 2016). That study also found downstream export to have a relatively low Si:C ratio, due to a lower proportion of empty diatom frustules, with over 40% of the sinking POC associated with diatom resting spores. The extensive summer blooms around South Georgia also deplete dissolved inorganic carbon (DIC), acting as a strong sink for atmospheric $CO_2$, although this is countered in the winter due to strong mixing (Jones et al., 2012).

Despite some retention over the continental shelf of South Georgia, the high productivity of phytoplankton is widespread, extending into deep waters to the north of the island and to the PF (>8 mg chlorophyll *a* m$^{-3}$, Fryxell et al., 1979; El-Sayed and Weber, 1982; Whitehouse et al., 1996b, 2000; Korb et al., 2004, 2008; Korb and Whitehouse, 2004; Hoppe et al., 2017). Such high productivity can also be sustained for thousands of kilometres downstream (Korb et al., 2004), in part due to the supply of nutrients by meandering mesoscale structures that form during interactions between the oceanic fronts and topographic features (Smith et al., 2010; Jones et al., 2017). Indeed, the large spatial extent and long growth season (Korb et al., 2004;

Hoppe et al., 2017) mean that the South Georgia blooms are associated with the strongest predicted carbon sink in the Southern Ocean (Schlitzer, 2002).


In this paper we present a multi-year nutrient dataset along with concurrent temperature and salinity measurements. We detail methods, consider preliminary data evaluation, and summarise previous data management and utilisation. In addition to data availability, we consider their potential use and also catalogue supporting cruise report information.

## 2 Methods

The present macronutrient database has been collated from direct measurements made by BAS scientists (principally MJW) as part of sampling campaigns conducted from scientific research vessels to the respective regions of the Southern Ocean. Samples were collected via two principal collection methods: CTD (conductivity, temperature, depth) rosette water bottles or, additionally on the RRS James Clark Ross, the ship's non-toxic seawater supply while underway. Initial measurements were made at sea and concentrations calculated post-cruise. These data were initially published in a number of different sources

which have been compiled as part of the present exercise (see Whitehouse et al., 2022). Furthermore, we have carried out additional quality control of these measurements and added supplementary data fields to the database that we are now making accessible externally (Whitehouse et al., 2022).

### 2.1 Sampling sites

The current dataset was collected between 1980 and 2009 (Table 1).


**Table 1. Cruise, duration, region and nutrients measured at CTD stations (●) and during transects (●). *MEB - Maurice Ewing Bank transect between 48°S, 44°W and 54°S, 39°W. Concurrent temperature and salinity values are presented for all CTD profiles and for cruises JR161, JR177 and JR200 transect measurements.**

| Cruise | From | To | Region | $Si(OH)_4$-Si | $PO_4$-P | $NO_3$-N | $NH_4$-N | $NO_2$-N |
|--------|------|-----|--------|---------------|----------|----------|----------|----------|
| JB03 | Nov-81 | Jan-82 | South Georgia | ● | ● | ● | | |
| JB04 | Jul-83 | Oct-83 | South Georgia | ● | ● | ● | | ● |
| JB05 | Jan-85 | Feb-85 | Bransfield Strait | ● | ● | ● | ● | ● |
| JB06 | Dec-85 | Jan-86 | South Georgia | ● | ● | ● | ● | ● |
| JB08 | Jan-88 | Mar-88 | South Georgia/Bransfield Strait | ● | ● | ● | ● | ● |
| JB10 | Jan-90 | Feb-90 | South Georgia | ● | ● | ● | ● | |
| JR02 | Nov-92 | Dec-92 | Bellingshausen Sea | ● | ● | ● | ● | ● |
| JR06 | Jan-94 | Feb-94 | South Georgia inc. MEB* | ● | ● | ● | ● | ● |
| JR11 | Jan-96 | Jan-96 | South Georgia inc. MEB* | ● | ● | ● | ● | ● |
| JR17 | Dec-96 | Jan-97 | South Georgia inc. MEB* | ● ● | ● ● | ● ● | ● ● | ● ● |

| JR25 | Oct-97 | Nov-97 | South Georgia inc. MEB* | ● ● | ● ● | ● ● | ● ● | ● ● |
| JR28 | Jan-98 | Feb-98 | South Georgia | ● ● | ● ● | ● ● | ● ● | ● ● |
| JR38 | Dec-98 | Jan-99 | South Georgia | ● ● | ● ● | ● ● | ● ● | ● ● |
| JR57 | Dec-00 | Jan-01 | South Georgia | ● ● | ● ● | ● ● | ● ● | ● ● |
| JR70 | Jan-02 | Feb-02 | South Georgia | ● ● | ● ● | ● ● | ● ● | ● ● |
| JR82 | Jan-03 | Feb-03 | Scotia Sea and South Georgia | ● ● | ● ● | ● ● | ● ● | ● ● |
| JR116 | Dec-04 | Jan-05 | South Georgia | ● | ● | ● | ● | |
| JR161 | Oct-06 | Nov-06 | Scotia Sea and South Georgia | ● ● | ● ● | ● ● | ● ● | |
| JR177 | Jan-08 | Feb-08 | Scotia Sea and South Georgia | ● ● | ● ● | ● ● | ● ● | |
| JR200 | Mar-09 | Apr-09 | Scotia Sea and South Georgia | ● ● | ● ● | ● ● | ● ● | |

Measurements were made predominantly around the island of South Georgia and across the wider Scotia Sea with additional sampling to the west of the Antarctic Peninsula and in the Bellingshausen Sea (Fig. 1a and 1b).

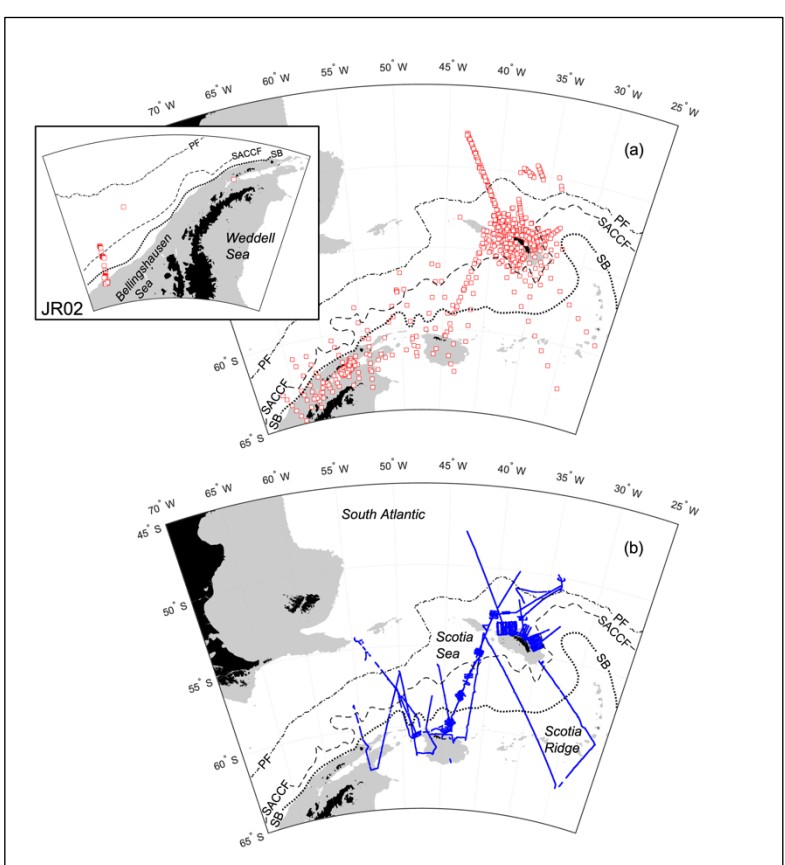

**Figure 1. Sample locations of (a) CTD vertical profiles with inset to show stations sampled in the Bellingshausen Sea during cruise**
**JR02; (b) transect data points. Southern Ocean fronts are shown in each panel: PF, Polar Front; SACCF, Southern Antarctic Circumpolar Current Front; and SB, Southern Boundary of the Antarctic Circumpolar Current (Orsi et al., 1995; Moore et al.,**

1999; Thorpe et al., 2002; Orsi and Harris, 2019). Pale grey areas show water depths <1000 m (Amante and Eakins, 2009; NOAA National Geophysical Data Center 2009). Coastline following GSHHG, version 2.3.7.

During nearly 30 years of sampling, some sites were repeatedly visited. Two monitoring areas termed the Eastern Core Box (ECB) and the Western Core Box (WCB) to the north of South Georgia have been studied intensively with a set of repeat transects oriented across the northeastern and northwestern shelf break to sample on- and off-shelf waters (Fig. 2a).

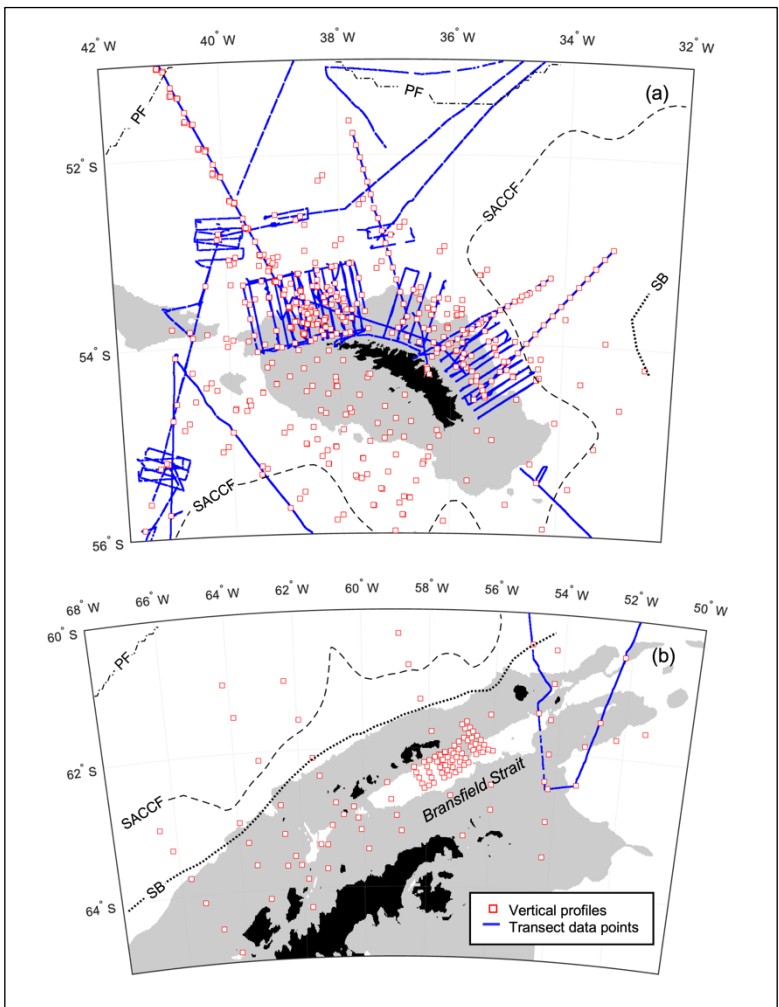

**Figure 2. Smaller-scale detail of sampling around (a) South Georgia; (b) Antarctic Peninsula. Southern Ocean fronts, pale grey areas and coastline are as in Figure 1.**

A series of stations spanning >700 km from the western end of South Georgia to the northwest crossed the PF and a number of bathymetric features including the Maurice Ewing Bank (50.667°S, 43.5°W) after which the transect was named. Three

cruises crossed the Scotia Sea from south to north in consecutive years and covered spring, summer and autumn periods (see Tarling et al., 2012). These cruises along with the Core Box surveys to the north of South Georgia included extensive underway sampling (see Section 2.2), while surveying near the Antarctic Peninsula comprised predominantly station grids (Fig. 2b).

## 2.2 Chemistry instrumentation and methods

In 1979, a segmented-flow analyser (SFA) was built in-house to analyse macronutrient measurements for the BAS offshore
programme. It was based on Chemlab colorimeters and Ismatec proportioning pumps (Whitehouse and Woodley, 1987). Originally, data were logged to paper chart and processed manually. In the mid-1980s, data extraction was automated and processing was managed with a digitising tablet and associated PC (Woodley, 1989). During 1993, the analyser was completely remodelled around Technicon colorimeters, the Ismatec proportioning pumps were updated, the chemistry manifolds were re-built and data were logged to a PC (Whitehouse, 1997). Data acquisition and subsequent processing was achieved with custom-
built software (see Whitehouse and Preston, 1997).

All data were collected during BAS or other Natural Environment Research Council cruises aboard the RRS John Biscoe (cruises prefixed JB) or the RRS James Clark Ross (cruises prefixed JR). The macronutrients analysed are described fully in Whitehouse and Woodley (1987). They agree with the internationally accepted conventions for silicic acid ($Si(OH)_4$–Si),
orthophosphate ($PO_4$–P), nitrate ($NO_3$–N), nitrite ($NO_2$–N), and ammonium ($NH_4$–N). All concentrations are expressed as mmol m$^{-3}$. A statistical analysis of the chemistry methods was documented in Whitehouse and Woodley (1987) but we present a summary for this analysis in Table 2.

**Table 2. Replication and limits of detection measured for chemistry methods (Whitehouse and Woodley, 1987).**

|  | $Si(OH)_4$-Si | $PO_4$-P | $NO_3$-N | $NH_4$-N | $NO_2$-N |
|---|---|---|---|---|---|
| Replication ($n$) | 9 | 20 | 12 | 9 | 12 |
| Concentration (mmol m$^{-3}$) | 71 | 3.2 | 0.7 | 3.6 | 8.9 |
| Standard deviation of replicates (% of mean) | 0.14 | 0.47 | 0.75 | 0.42 | 0.8 |
| Limit of detection (mmol m$^{-3}$) | 0.28 | 0.05 | 0.28 | 0.01 | 0.01 |

Data were collected by means of two sampling methods. First of all, sub-samples were taken from CTD rosette water bottles that provide the conventional vertical profiles. Water samples were taken from CTD upcasts, in which bottles were fired at depths ranging from 5500 m to 0.5 m. Secondly, samples were analysed continuously from the ship's non-toxic seawater supply while the vessel was underway to provide high-resolution horizontal coverage of near-surface conditions during
transects. Here we document further details of the sample handling regime.

The ship's non-toxic seawater inlet was at ~6-7m and forward of the bows to avoid contamination. The time-lag between seawater entering the ship's non-toxic inlet and reaching the chemistry laboratory was ~60 seconds (S. Wright, Deck Engineer, pers. comm., 31 December 1996). On arrival at the chemistry laboratory, the stream of seawater passed through a tangential flow filter block (Morris et al., 1978) fitted with a filter membrane (pre-1990 Whatman GF/C, pore size 1.2 µm, and then a mixed ester Whatman WME, pore size 0.45 µm) (Fig. 3).

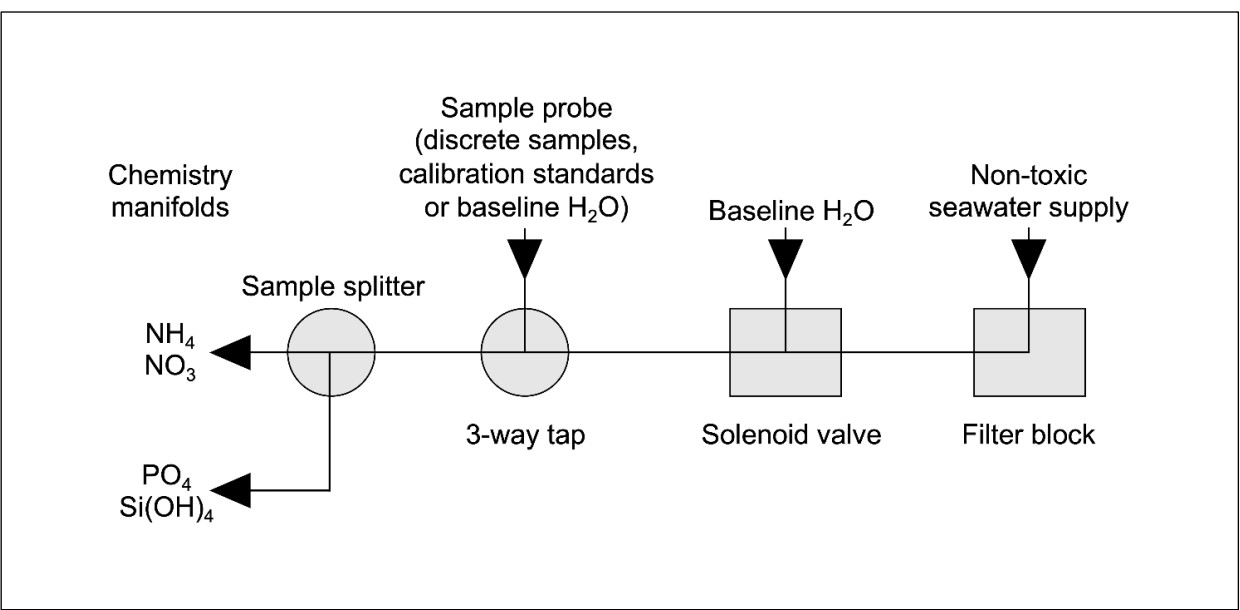

**Figure 3. Sample handling schematic for flow of samples, standards and baseline blanks between ship's non-toxic seawater supply and the chemistry manifolds of the segmented-flow analyser.**

A sub-sample was then pumped through a solenoid valve controlled with an electric timer. Typically, a 5 min baseline check was automatically introduced once per hour while the ship was underway and horizontal profiles were being measured. The sample stream then passed through a 3-way tap before arriving at a sample stream splitter followed by introduction to the chemistry manifolds. The 3-way tap allowed manual interruption of the sample stream to introduce discrete samples from the CTD water bottles along with calibration standards and blank solutions for the calculation of baselines, refractive index compensation and flow-through times.

The time-lag between the sample entering the laboratory and passing through the analyser to the detector varied between the different nutrient analyses depending on the complexity of the chemistries used. Therefore, underway measurements were time and date stamped at the analyser's detector by referencing the ship's central clock to enable integration with navigation and other underway measurements such as physical oceanography and phytoplankton-related information (see Whitehouse and Preston, 1997). The underway data were originally logged at a 10 second interval. However, given the time-lag between

seawater entering the ship's non-toxic inlet and reaching the chemistry laboratory (~60 seconds), the detector's output was
averaged over one-minute intervals during post-cruise processing to give a more realistic sampling sensitivity.

### 2.3 Temperature and conductivity measurements

Temperature and conductivity measurements were taken during CTD vertical profiles and with a thermosalinograph as part of the underway system that sampled from the ship's non-toxic seawater supply. During most cruises, the vertical profile conductivity measurements were calibrated against samples drawn from the CTD water bottles. Temperature and salinity data

from the profile upcast were averaged onto the depth and time of the bottle firing for most nutrient samples taken from the CTD water bottles but, where this was not possible, the downcast profile data were binned onto the depth ($\pm$2 metres) of the nutrient sample. Further details are provided in the respective cruise reports (Appendix 1). For this dataset, the underway temperature and conductivity measurements have not been calibrated against sample measurements but were averaged over a 60 second interval to smooth the data.

### 2.4 Dataset compilation

The compilation of the dataset is summarised in Fig. 4.

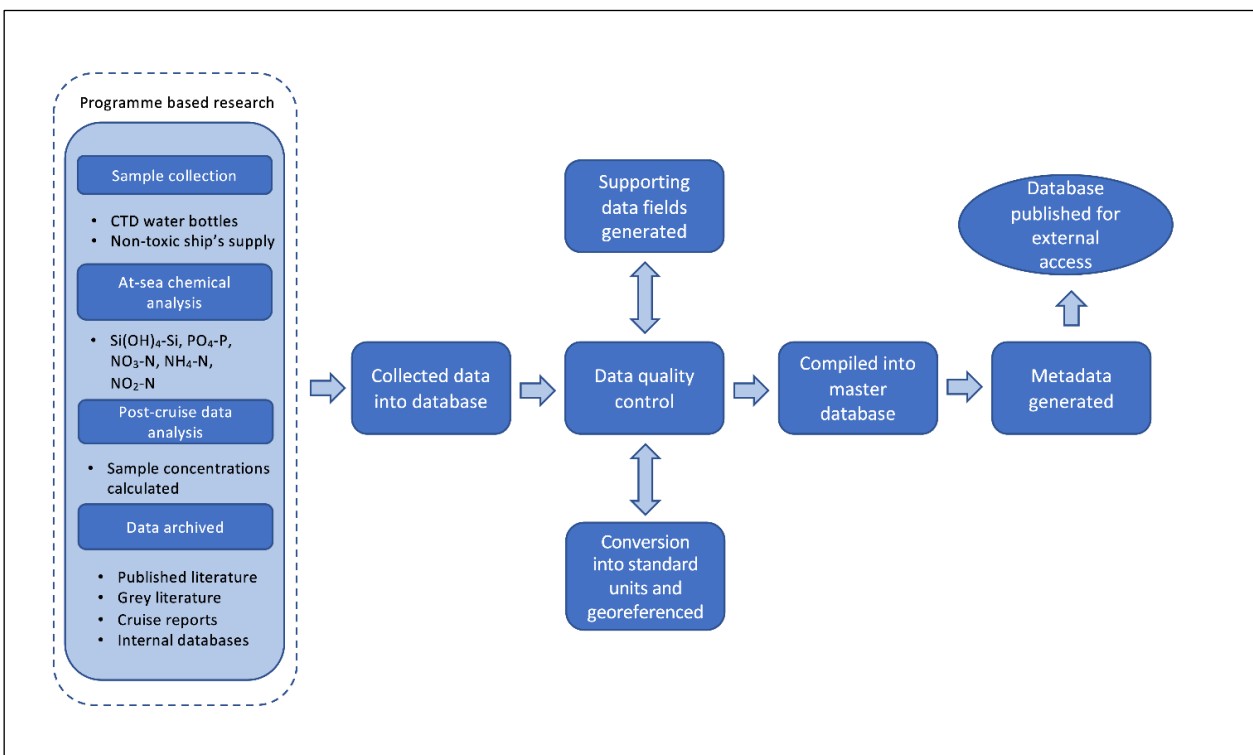

**Figure 4. Flowchart of dataset compilation including sample collection, data processing and utilisation.**

CTD water bottle data were compiled with ship, ship identification information, cruise number, event number, geographic location and depth. Temperature and salinity data were extracted and matched with profile nutrient data from each cruise using latitude and depth as primary selector variables, with event numbers and timestamps used as secondary checks.

The ship's non-toxic seawater supply data were compiled with ship identification information, cruise number, geographic location, temperature and salinity, and each timestamp reformatted as a date vector, a serial date number, and a datetime string (DD-MMM-YYYYThh:mm:ssZ). The serial date numbers were ordered temporally, converted into a datetime string (DD-MMM-YYYYThh:mm:ssZ), and converted to a table with minute intervals and corresponding arithmetic means of concentrations of each nutrient, temperature and salinity.

The profile and underway sample data have been deposited at the UK Polar Data Centre and are publicly available in NetCDF and CSV formats (Whitehouse et al., 2022). All available cruise reports (17 in total) that provide cruise and individual projects as well as additional background information on the current dataset have been compiled (Appendix 1).

## 3. Results

**3.1 Data evaluation**

In all, ~10,000 CTD water bottles were sampled and analysed, and ~900,000 one-minute bins of data were recorded from the underway sampling over the period 1980-2009. As with other high latitude marine sampling projects, logistics and sea ice conditions dictated when surveys could be conducted. January was the most frequent month of collection although sampling occurred in all months apart from May and June (Fig. 5).


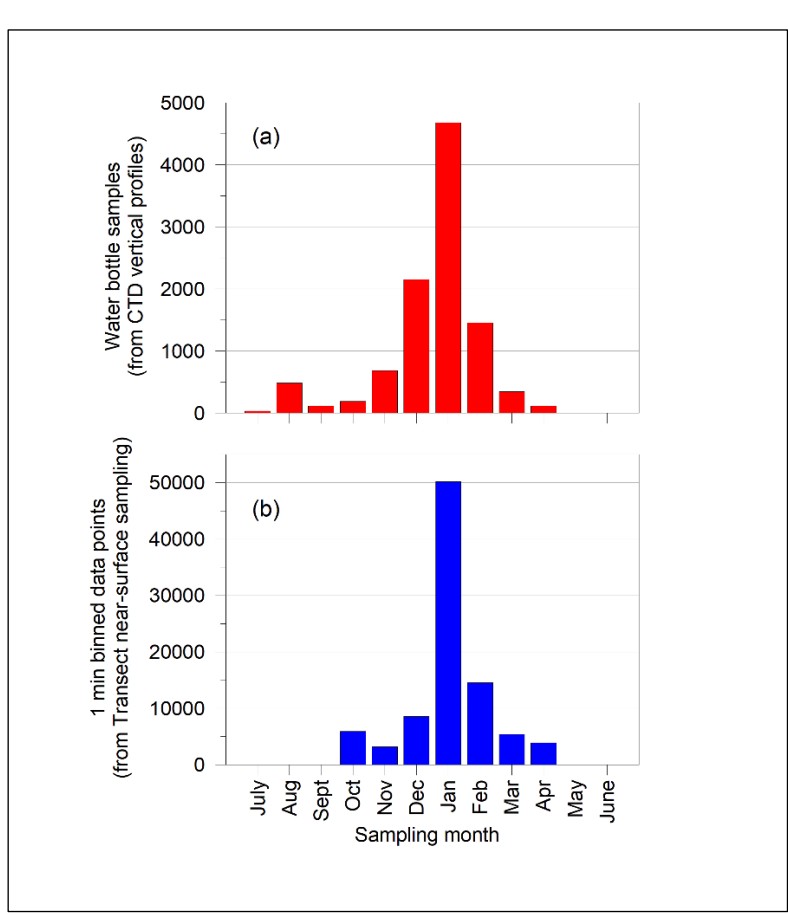

**Figure 5. Monthly totals for nutrient samples from CTD water bottles and transect data points.**

Oceanographically, the profile data spanned the southern boundary of the ACC (SB), SACCF, PF and the Subantarctic Front
(SAF) to the north of the PF (Fig. 6a). Antarctic Surface Water, Polar Frontal Zone surface water, and shelf waters were evident
in the underway data (Fig. 6b).

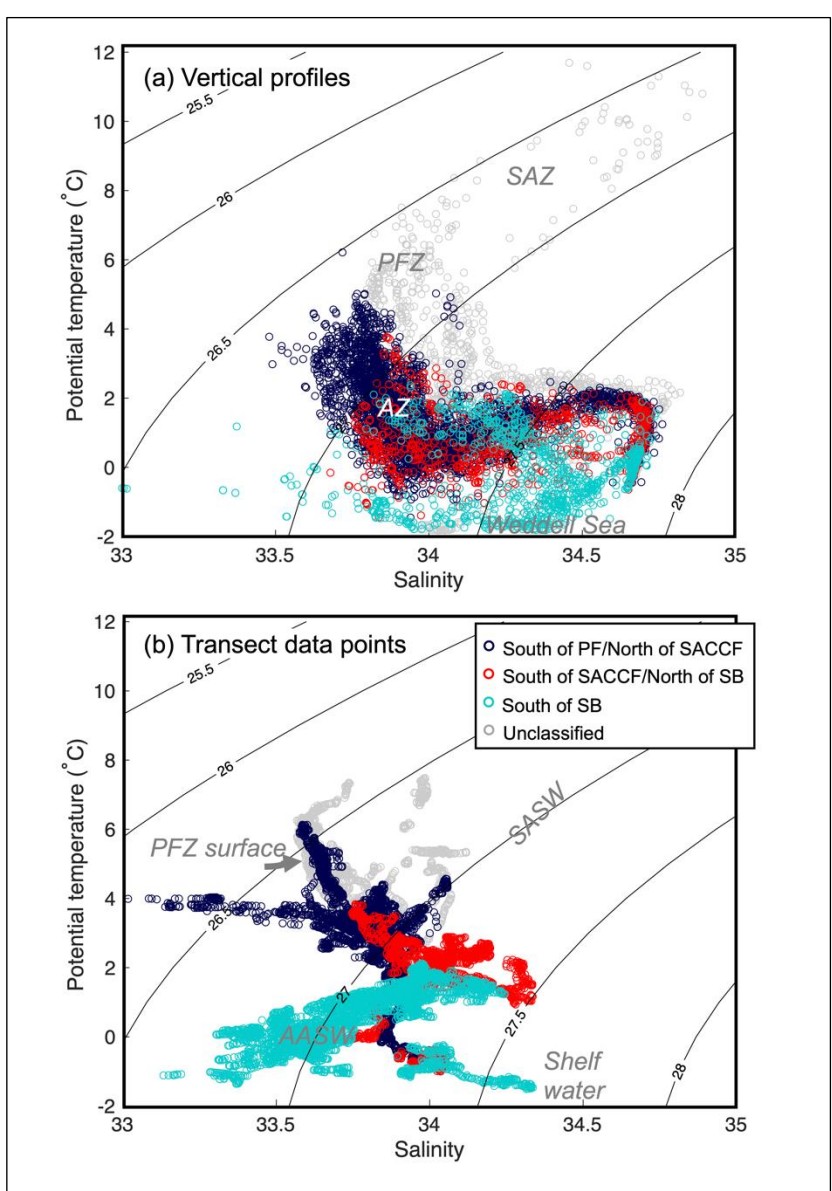

**Figure 6. Potential temperature-salinity plots for (a) CTD vertical profiles and (b) transect data points. Data have been colour coded**
**according to their location relative to the long-term mean positions of the PF, SACCF and SB (see Fig. 1). SAZ – Subantarctic Zone,**
**PFZ – Polar Frontal Zone, SASW – Subantarctic Surface Water, AASW – Antarctic Surface Water.**

On repeated transects to the north of South Georgia (cruises JR17, JR28 and JR38) and across the Scotia Sea (JR161, JR177
and JR200), near-surface nutrient values (0–7 m) were compared with a mean value derived from CTD water bottle data (0–
50 m) to assess whether surface values were representative of deeper parts of the water column (Table 3).

**Table 3. Relationship between mean surface (0 to 7 m) nutrient data with those averaged over 0 to 50 m using linear regressions. All available data from 6 cruises were examined during 2 studies: (a) to the north of South Georgia with 50% of stations on-shelf (Whitehouse et al., 2009); (b) repeated transects across the Scotia Sea sampling deep oceanic waters (Whitehouse et al., 2012).**


| Parameter | (a) Cruises JR17, JR28, JR38 | | | (b) Cruises JR161, JR177, JR200 | | |
|---|---|---|---|---|---|---|
| | n | $R^2$ | p | n | $R^2$ | p |
| $SiOH_4$-Si | 29 | 98% | <0.001 | 31 | 100% | <0.001 |
| $PO_4$-P | 31 | 94% | <0.001 | 29 | 93% | <0.001 |
| $NO_3$-N | 31 | 96% | <0.001 | 31 | 98% | <0.001 |
| $NH_4$-N | 31 | 87% | <0.001 | 29 | 69% | <0.001 |

Linear regressions for all comparisons were highly significant ($p < 0.001$). For $Si(OH)_4$-Si, $PO_4$-P, and $NO_3$-N, $R^2$ values were
$\geq 93\%$. For $NH_4$-N, $R^2$ values were lower and a difference was observed between the South Georgia cruises (87%) and those
in the Scotia Sea (69%). This was doubtless due to the different water column characteristics sampled during the two studies.
The South Georgia cruises include many on-shelf stations to the north of the island where $NH_4$-N is biologically generated and
constrained bathymetrically to the surface waters. Whereas for the offshore Scotia Sea stations, $NH_4$-N concentrations were
relatively low, pycnoclines were deeper and the upper mixed-layer frequently extended below 50 m (Korb and Whitehouse,
2004; Korb et al., 2012).


### 3.2 Regionality

To illustrate regional variation within the dataset we subsampled austral summer, near-surface (0–50 m) profile data from eight
areas in distinct oceanographic regimes (Fig. 7, Table 4).

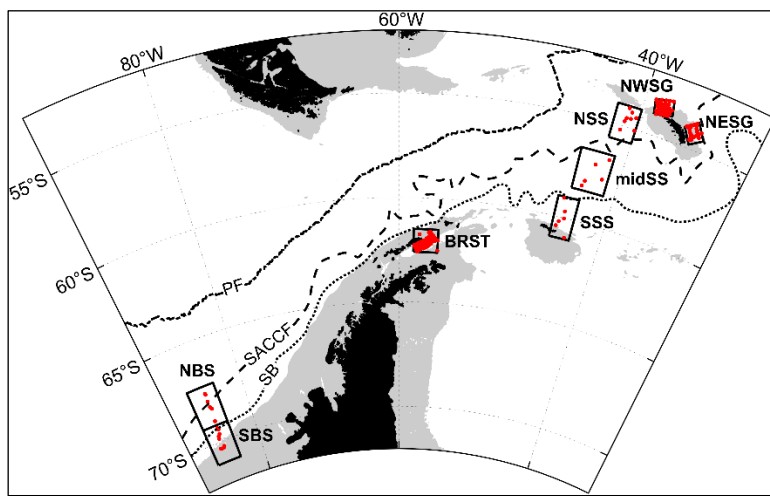


**Figure 7. Areas subsampled to illustrate regional variation: NWSG, northwest South Georgia; NESG, northeast South Georgia; NSS, north Scotia Sea; midSS, mid-Scotia Sea; SSS, south Scotia Sea; BRST, Bransfield Strait; NBS, north Bellingshausen Sea; and SBS, south Bellingshausen Sea. Red dots mark sample locations included in the analyses. Southern Ocean fronts and bathymetry are shown as in Fig. 1.**


**Table 4. Details of the profile samples used to derive the austral summer properties for 0-50 m (Fig. 8). Refer to Fig. 7 for locations. Number of samples refers to the number of Niskin bottles for which at least one inorganic nutrient was analysed for the given location, months and depth range.**

| Location | Month(s) | Year(s) | Number of stations | Number of samples at 0-50 m |
|---|---|---|---|---|
| NWSG | December, January | 1981, 1985, 1986, 1990, 1994, 1996, 1997, 1998, 1999, 2001, 2002, 2005 | 120 | 456 |
| NESG | December, January | 1981, 1982, 1996, 1998, 2000, 2001, 2002 | 49 | 173 |
| NSS | December, January | 1981, 1988, 2003, 2008 | 8 | 35 |
| midSS | January | 2003, 2008 | 5 | 18 |
| SSS | January | 2003, 2008 | 7 | 31 |
| BRST | January, February | 1985 | 105 | 160 |
| NBS | December | 1992 | 8 | 39 |
| SBS | November, December | 1992 | 12 | 66 |


### 3.2.1 South Georgia

Early descriptions of silicate and phosphate distributions at South Georgia were presented by Clowes (1938), but modern techniques have expanded spatial and temporal sampling and allowed the analysis of nutrients such as nitrate, nitrite, and
ammonium in addition to silicate and phosphate.

To the northwest of South Georgia, the warmest and most productive waters with regards phytoplankton are found. Indeed, the area is predicted to be the strongest carbon sink in the Southern Ocean (Schlitzer, 2002), and phytoplankton blooms can extend >2000 km downstream to the east (de Baar, et al., 1995; Korb, et al., 2004). This is the most comprehensively sampled
of the eight areas presented in this analysis (Fig. 7, Table 4,). Surface temperatures average ~3°C in summer (Fig. 8) and

macronutrients are generally plentiful apart from silicate that can be depleted to limiting concentrations. In addition, marine life is abundant around the island supplying multiple sources of regenerated, reduced nitrogen.

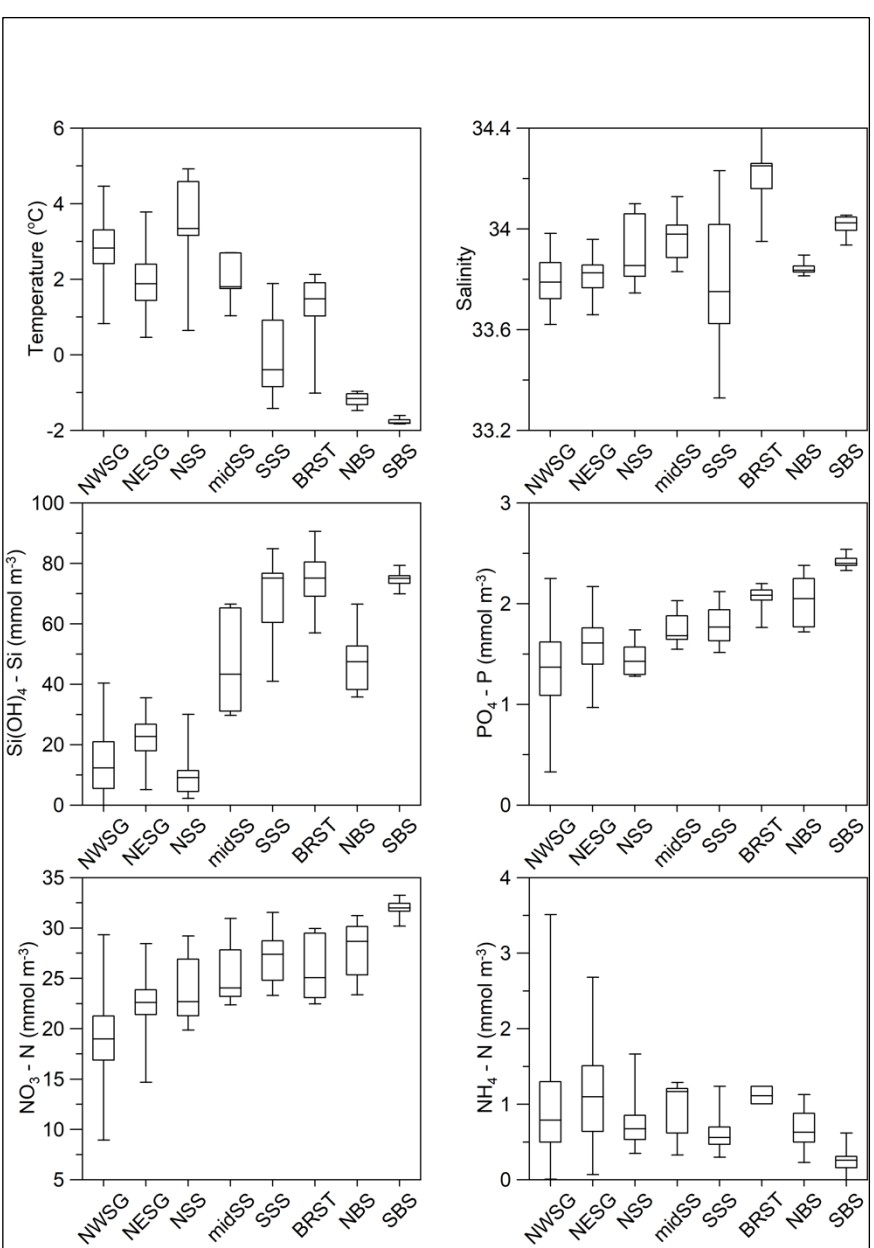

 **Figure 8. Regional variability in physical and chemical near-surface (0-50 m) ocean properties for austral summer profile data. The box and whisker plots comprise a box that is defined by the region's lower and upper quartiles, the line in the center of the box is the median, and the whiskers indicate the extreme values (minimum and maximum). Refer to Fig. 7 for sample locations and to Table 4 for temporal coverage and numbers of data points.**

Waters to the northeast of South Georgia are influenced by the SACCF and are cooler. Macronutrient concentrations are higher than to the west of the island but even so, silicate levels are low relative to the Southern Ocean as a whole. By contrast, ammonium concentrations tend to be higher and this has been linked to nitrogen regenerated by an abundance of euphausiids and copepods in this region (Atkinson and Whitehouse, 2000, 2001). Historically, Antarctic krill *Euphausia superba* and krill predator abundance were highest in this region, although whale populations were demolished in the early 20th century

(Headland, 1984), and krill abundance around the island has declined in recent years (Atkinson et al., 2022).

A compilation of data collected before the mid-1990s (including part of the current dataset) presented seasonal trends for macronutrients and considered their relationship with physical oceanography, phytoplankton use, and interannual variability (Whitehouse et al., 1996a). Further subsets of the current dataset have been used in a comparison of conditions around the

island and contrasting the east versus west (Whitehouse et al., 1993; Whitehouse et al., 1999); in considering nutrient dynamics around the island's shelf break (Whitehouse et al., 2008), and the Maurice Ewing Bank transect (Whitehouse et al., 1996b; Whitehouse et al., 2000). Atkinson's South Georgia review (2001) sets nutrient characteristics in an ecological context and the role of macronutrients in phytoplankton dynamics has been explored in several studies (e.g., Atkinson and Whitehouse, 2000; Whitehouse et al., 2009, 2011). Ocean warming (by ~1°C over the last century) has also been documented for the island

(Whitehouse et al., 2008b).

### 3.2.2 Scotia Sea

Phytoplankton productivity in the Scotia Sea is high relative to the Southern Ocean as a whole, but it is subject to strong

temporal and spatial variability, largely associated with oceanographic gradients across the region's frontal zones (see Tarling et al., 2012). This variability is evident in the physical properties of the water: the mid-Scotia Sea is cooler than the northern Scotia Sea as it lies south of SACCF, and the southern Scotia Sea is colder again being south of the SB. The southern Scotia Sea has the largest range in summer salinity, possibly due to interannual variability in sea ice extent in this region (Fig. 8).

This summer variability is also evident in macronutrient concentrations. Similar to the northwest of South Georgia, the north Scotia Sea has relatively warm surface waters due to being north of the SACCF and silicate concentrations may be limiting for diatom growth (Fig. 8). Also, reduced nitrate and elevated ammonium concentrations indicate both iron availability and regeneration of reduced nitrogen.

Silicate concentrations increase into the mid-Scotia Sea, and again into the southern Scotia Sea region (Fig. 8), reflecting the biogeochemical gradients across the SACCF and SB. Nitrate and phosphate concentrations remain relatively high in the mid-Scotia Sea and southern Scotia Sea regions although ammonium levels, especially in the mid-Scotia Sea, indicate a degree of

nitrogen regeneration. Multi-seasonal macronutrient concentration variability is documented by Whitehouse et al. (2012), as are their relationships with primary production (Korb et al., 2005; Korb et al., 2012).


### 3.2.3 Bransfield Strait and the Bellingshausen Sea

Many of the stations sampled in the Bransfield Strait comprise the BAS contribution to SIBEX (the Second International BIOMASS Experiment) (El Sayed, 1994). As far as we know the macronutrient measurements have only been used partially

in relation to phytoplankton distribution (Heywood and Priddle, 1987) and physiology (Owens et al., 1991).

Sampling in this region took place primarily in January and February. Cool surface waters and relatively high macronutrient concentrations are typical of the southern region of the Southern Ocean (Fig. 8). However, elevated ammonium concentrations have been measured here associated with a variety of phytoplankton communities (Owens et al., 1991). In early February 1985,

silicate depletion correlated with a patch of high phytoplankton biomass off King George Island, South Shetland Islands. The patch remained stationary over a period of at least 11 days and was centred on an eddy at the apex of a tight meander formed by water passing around the eastern end of King George Island and being turned back immediately by the strong northeastern flow of water within the Bransfield Strait (Heywood and Priddle, 1987).

Unlike the previous regions, the Bellingshausen Sea samples were collected during late spring and early summer (November and December) as part of a marginal ice zone survey. This study, entitled the STERNA cruise, was a component of Southern Ocean JGOFS (Joint Global Ocean Flux Study, Turner et al., 1995).

These data, in contrast to those from the other regions, document the initiation of a phytoplankton bloom at the break-out of

the winter's pack ice. The southern stations show typical winter conditions with cold, high nutrient waters apparent (Fig. 8). To the north, the major reduction in silicate concentrations is related to latitudinal hydrographic change. However, the waters have begun to warm, and there are indications of nutrient drawdown. Macronutrient variability and uptake are described by Whitehouse et al. (1995).

### 4. Data availability

The profile and underway sample data have been deposited at the UK Polar Data Centre and are publicly available in NetCDF and CSV formats from https://doi.org/10.5285/4014370F-8EB2-492B-A5F3-6DC68BF12C1E (Whitehouse et al., 2022).

**5. Code availability**

This paper does not report original code.

**6. Conclusions and potential use of the dataset**

We have presented here a new macronutrient database, including depth profiles and continuous underway surface measurements of silicic acid, phosphate, nitrate, ammonium, and nitrite, along with co-located temperature and salinity. Data were collected throughout the year from a variety of biological "hotspots" within the Southern Ocean, including South Georgia and the wider Scotia Sea, the Western Antarctic Peninsula, and the Bellingshausen Sea. One of the key challenges in understanding carbon cycling in the Southern Ocean is disentangling long-term responses from significant spatial and temporal

variability in physical and biogeochemical parameters. As such, there is a critical need for regional long-term observational datasets that are openly accessible to generate a better mechanistic understanding of the drivers of primary production. The new data product described here provides an unprecedented view of biogeochemical cycling in biologically productive regions of the Southern Ocean across a critical period in recent climate history, and illustrates the importance of FAIR (Findable, Accessible, Interoperable, and Reusable) sharing of scientifically valuable observational datasets.

**7. Appendix 1**

Further information is available for most of the cruises during which data for the current database were collected. This includes the specific aim of each cruise, methodology and preliminary results.

Cruise reports are listed in chronological order:


Heywood, R. B.: RRS John Biscoe Cruise JB04: Offshore Biological Programme, BODC Cruise Inventory, 32 pp., https://www.bodc.ac.uk/resources/inventories/cruise_inventory/reports/john_biscoe4_83.pdf, 1983.

Priddle, J.: RRS John Biscoe Cruise JB08: South Georgia and Bransfield Strait Marine Biology, BODC Cruise Inventory, 52

pp., https://www.bodc.ac.uk/resources/inventories/cruise_inventory/reports/john_biscoe08.pdf, 1988.

Watkins, J. L., and Priddle, J.: RRS John Biscoe Cruise JB10: South Georgia Marine Biology, BODC Cruise Inventory, 12 pp, https://www.bodc.ac.uk/resources/inventories/cruise_inventory/reports/john_biscoe10_90leg1_2.pdf, 1990.

Owens, N. J. P.: RRS James Clark Ross Cruise JR02: Biogeochemical Ocean Flux Study, BODC Cruise Inventory, 245 pp, https://www.bodc.ac.uk/resources/inventories/cruise_inventory/reports/jr02_92.pdf, 1992.

White, M.: RRS James Clark Ross Cruise JR06: South Georgia and South Orkneys Marine Biology (Predator/Prey cruise), BODC Cruise Inventory, 218 pp, https://www.bodc.ac.uk/resources/inventories/cruise_inventory/reports/jr06.pdf, 1994.


Priddle, J., and Swanson, C.: RRS James Clark Ross Cruise JR11: Marine Life Sciences Pelagic Ecosystem Studies, BAS Cruise Archive, 46 pp, 1996.

Priddle, J.: RRS James Clark Ross Cruise JR17: BAS Marine Life Sciences, BAS Cruise Archive, 44 pp, 1996.


Priddle, J.: RRS James Clark Ross Cruise JR25: Spring Processes, BAS Cruise Archive, 116 pp, 1997.

Murphy, E. J.: RRS James Clark Ross Cruise JR28: Variability of the South Georgia Marine Ecosystem, BODC Cruise Inventory, 160 pp, https://www.bodc.ac.uk/resources/inventories/cruise_inventory/reports/jr028.pdf, 1998.


Everson, I.: RRS James Clark Ross Cruise JR38: Variability of the South Georgia Marine Ecosystem, BODC Cruise Inventory, 55 pp, https://www.bodc.ac.uk/resources/inventories/cruise_inventory/reports/jr38.pdf. 1999.

Ward, P.: RRS James Clark Ross Cruise JR57: Variability in the Southern Ocean, BODC Cruise Inventory, 77 pp,
https://www.bodc.ac.uk/resources/inventories/cruise_inventory/reports/jr57.pdf, 2001.

Ward, P.: RRS James Clark Ross Cruise JR70: Flux and Marine Production Experiment, BODC Cruise Inventory, 155 pp, https://www.bodc.ac.uk/resources/inventories/cruise_inventory/reports/jr70.pdf, 2002.

Atkinson, A.: RRS James Clark Ross Cruise JR82: Large scale distribution in the Scotia Sea, BODC Cruise Inventory, 164 pp, https://www.bodc.ac.uk/resources/inventories/cruise_inventory/reports/jr82.pdf, 2003.

Ward, P.: RRS James Clark Ross Cruise JR116: A study of community structure and production along the southern shelf of South Georgia and links with the Georgia Basin, BODC Cruise Inventory, 127 pp,
https://www.bodc.ac.uk/resources/inventories/cruise_inventory/reports/jr116.pdf, 2005.

Shreeve, R.: RRS James Clark Ross Cruise JR161: A study of pelagic marine food web interactions and condition factors of zooplankton across the Scotia Sea, BODC Cruise Inventory, 324 pp, https://www.bodc.ac.uk/resources/inventories/cruise_inventory/reports/jr161.pdf, 2006.


Tarling, G.: RRS James Clark Ross Cruise JR177: Life cycles and trophic interactions of the Scotia Sea pelagic community: from ice-edge to Polar Front, BODC Cruise Inventory, 321 pp, https://www.bodc.ac.uk/resources/inventories/cruise_inventory/reports/jr177.pdf, 2008.

Korb, R.: RRS James Clark Ross Cruise JR200: Life cycles and trophic interactions of the Scotia Sea pelagic community: from the South Orkneys to the Polar Front, BODC Cruise Inventory, 205 pp, https://www.bodc.ac.uk/resources/inventories/cruise_inventory/reports/jr200.pdf, 2009.

For cruises JR11, JR17 and JR25, pdf files can be requested from the Polar Data Centre at PDCServiceDesk@bas.ac.uk.

Cruise reports are not available for JB03, JB05 or JB06.

**8. Author contributions**

MJW, KRH, GAT, SET and PTH conceived the data paper and wrote the manuscript. All authors commented on the paper and contributed to the quality check.

**9. Competing interests**

The authors declare that they have no conflict of interest.

**10. Acknowledgements**

We are most grateful to the officers and crew of the RRS John Biscoe and the RRS James Clark Ross, along with the cruises' Principal Scientists. We are also indebted to Vince Woodley, Julian Priddle, Beki Korb and Min Gordon for their assistance
with chemical and data analysis, and to the numerous scientists and technicians for their assistance with CTD and underway sampling and calibration of temperature and salinity data. MJW would also like to thank Barry Heywood, Julian Priddle and Pete Ward for their advice and guidance over many years. Finally, we are grateful to an anonymous referee and Dr. Elizabeth Jones for their constructive comments on an earlier manuscript draft.

**11. Financial support**

The contributions of KRH, GAT, SET and PTH were supported by the BIOPOLE National Capability Multicentre Round 2 funding from the Natural Environment Research Council (NE/W004933/1). GAT and SET were also supported by NC-ALI funding to the Ecosystems team at the British Antarctic Survey.

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
