# Peer review of "A database of marine macronutrient, temperature and salinity measurements made around the highly productive island of South Georgia, the Scotia Sea and the Antarctic Peninsula between 1980– 2009"

_Earth System Science Data, 2022_

## Author Response (AR3)

**Response to all referees' comments**

Here we provide a detailed point-by-point response to all referee comments regarding the manuscript "A database of marine macronutrient, temperature and salinity measurements made around the highly productive island of South Georgia, the Scotia Sea and the Antarctic Peninsula between 1980–2009" by Whitehouse et al.

**RC1: 'Comment on essd-2022-244', Anonymous Referee #1, 23 Aug 2022**

*The manuscript compiles macronutrient data in a region of high importance for the understanding of natural effects and climate change on physical, chemical and biological processes in the Southern Ocean, and the relevance of making these unique data available for use by the scientific community is appreciated.*

*I strongly agree with the publication of this robust dataset, however, I see that the authors could extract some more information from the compiled data before final publication, so that the scientific community can benefit even more.*

*In this sense, I missed a descriptive section on the macronutrients of the study region. I suggest the authors explore the TS-[nutrients] charts. And also, if possible, create climatological maps of these parameters, so that the scientific community (observers and modelers) can use a derived product for verifications, assessments and validations of models and/or other derived datasets (buoys etc.). Some good examples in ESSD can be found at: https://doi.org/10.5194/essd-8-15-2016; https://doi.org/10.5194/essd-13-671-2021. I understand that there may be temporal and spatial constraints to this approach, but from the information reported, at least the area around South Georgia Island for the summer period (DJF or JFM) could be considered.*

*Finally, the manuscript is well written and concise and have relevant figures/tables. The references are up-to-date. Thus, I think that the manuscript is worth of publication after a round of major review, focusing on the above comments/suggestions.*

The Reviewer strongly agrees with the publication of our dataset but feels we could extract more descriptive information from the compiled data. This is in line with the suggestions of Reviewer 2 and we have now added a new section to the paper (Section 3.2 Regionality) that provides statistical descriptions of the summer, near-surface properties of eight distinct oceanographic regimes within our sampling region together with references to previously published works that describe subsets of the data, as described more fully in our response to Reviewer 2, below. The Reviewer identifies the temporal and spatial constraints on creating climatological maps – although the data cover a 30-year period they also comprise multiple, regional datasets that cover >50° of longitude and >20° of latitude and do not lend themselves to wide-scale climatological mapping. However, the metadata provided in the dataset allow interested users to derive climatological maps that suit their

requirements in terms of spatial and temporal resolution. Our main aim of making these data available to the scientific community is to inspire fresh interpretations and integration with other datasets, and we thank the reviewer for their positive comments on our dataset and paper.

**RC2: 'Comment on essd-2022-244', Elizabeth Jones, 23 Aug 2022**

*General Comments*

*This study presents a compilation of macronutrient data collected during the period 1980-2009 from an important biological region in the Southern Ocean. The valuable and extensive data comprise vertical profiles and surface measurements and are presented in the context of distinguishing spatial and temporal variability from long-term change in a dynamic region. The manuscript is well written and accompanied by well-presented figures and key information is reported in tables. I recommend publication of the dataset and the manuscript. In the version of the manuscript reviewed here there are a number of minor revisions that can be made before proceeding with publication.*

*Specific Comments*

*The results section would greatly benefit from a description of the macronutrient variability with respect to the different water masses, fronts and regions, as described in the Introduction.*

Both Reviewers have suggested that the paper would be improved with the addition of more data description. In our response to Reviewer 1, we mentioned that a number of subsets of the dataset have already been published. These vary from seasonal considerations of the data around South Georgia and across the Scotia Sea, along with nutrient utilization by phytoplankton and physiological regeneration by zooplankton. Because of these previously published analyses, we feel it would be most beneficial (and efficient) to:

a. describe the regional variation in the surface waters associated with the full array of oceanographic regimes present in the study area

b. draw the readers' attention to data descriptions and uses published previously

c. detail additional pertinent information

We have changed the manuscript by including an analysis of near-surface (0 – 50 m) water properties during the austral summer within 8 regions: northwest South Georgia; northeast South Georgia; north Scotia Sea; mid-Scotia Sea; south Scotia Sea; Bransfield Strait; north Bellingshausen Sea; and south Bellingshausen Sea (see Section 3.2 Regionality). We show the geographic and oceanographic locations of these regions with a new figure (Fig. 7), and tabulate the number of samples in our analysis (number of sampling locations and the number of Niskin bottle samples for

these locations), along with months and years within which samples were collected (Table 4). We present the regional statistics (median, upper and lower quartile, minimum and maximum) in a new figure (Fig. 8). Our accompanying text details salient features in each region, highlights previous data descriptions and uses, and mentions additional, pertinent topics (e.g., ocean warming and the changing abundance of krill). We thank the reviewer for their positive comments on our paper.

Below we respond to RC2's specific points

*In the Methods section, are descriptions the same both of the vessels (JB and JCR)? For example, the depth of the underway seawater supply and time taken for seawater to pass through the ship.*

We have deleted "or the underway non-toxic ship's seawater supply" and inserted "or, additionally on the James Clark Ross, the ship's non-toxic seawater supply while underway". (lines 138 - 139)

*Usage of (a) and (b) in figures: in figure 1 and 2 the (a) and (b) have small boxes around and in figures 5 and 6 they have no boxes.*

We have deleted the boxes in Figs 1 & 2.

*Technical Corrections*

*Line 48 how do the eddies "… influence…" the region? One or two examples would complement the statement*

We have rewritten this sentence so that it now reads:

"There is some evidence, for example, that eddies from north of the PF input warmer water to the island and introduce physical structure to the water column that facilitate the resupply of nutrients from deeper to surface waters (Atkinson et al., 1990; Whitehouse et al., 1996b)." (lines 47 - 49)

*Line 76 insert 'marine environment' after 'South Georgia'*

We have rewritten this description so that it now reads:

"in the South Georgia marine environment (line 76)

*Line 107 insert 'phytoplankton' before 'bloom'*

Done (now line 110)

*Line 110-111 repetition of the phrase 'play a role'*

We have substituted "likely play a role in" with "influence" (now line 113)

*Line 111 is the word 'and' after 'nutrient cycling' a typo?*

We have now inserted "," after cycling (now line 114)

*Line 241 is the word 'ship' after 'compiled with' a typo?*

We have amended the sentence to now read:

"The ship's non-toxic seawater supply data were compiled with ship identification information, cruise number, geographic location, temperature and salinity, and each timestamp reformatted as a date vector, a serial date number, and a datetime string (DD-MMM-YYYYThh:mm:ssZ)." (lines 245 - 247)

**RC3: 'Comment on essd-2022-244', Anonymous Referee #1, 8 Nov 2022**

*The authors made some improvements during the review process and partially accepted some of my suggestions. Although I still think that additional exploration of the dataset could be performed to better benefit the community, the inclusion of Fig. 8 and the new "Regionality" section partially help on this. So, I think the paper can be accepted after some minor review: (i) include the geographical names used in the text in Fig. 1 (Weddell Sea, Scotia Sea, Bransfield Strait etc.). This will help readers not familiar with the region; (ii) avoid "isolated" phrases in the text for a more cohesive reading (e.g. lines 76-77, line 145, lines 151-152, lines 248-252, lines 359-360 etc.).*

We thank the reviewers for their positive comments on our dataset and paper.

We have added place names to figures 1 and 2 where room on the plot allowed. We have not labelled South Georgia or Antarctic Peninsula but their locations along with data points are the subject of Figure 2.

We have made the required changes to figure 6 so that all the colours can be differentiated in monochrome.

We have removed all the short one-sentence paragraphs. However, by complying with ESSD formatting requirements, some sentences have become isolated due to the insertion of figures and tables. These issues will be will be picked up and resolved at the copy-editing stage.